# POLAR PROTOTYPE NETWORKS

## ABSTRACT

This paper proposes a neural network for classification and regression, without the need to learn layout structures in the output space. Standard solutions such as softmax cross-entropy and mean squared error are effective but parametric, meaning that known inductive structures such as maximum margin separation and simplicity (Occam's Razor) need to be learned for the task at hand. Instead, we propose polar prototype networks, a class of networks that explicitly states the structure, *i.e.,* the layout, of the output. The structure is defined by polar prototypes, points on the hypersphere of the output space. For classification, each class is described by a single polar prototype and they are *a priori* distributed with maximal separation and equal shares on the hypersphere. Classes are assigned to prototypes randomly or based on semantic priors and training becomes a matter of minimizing angular distances between examples and their class prototypes. For regression, we show that training can be performed as a polar interpolation between two prototypes, arriving at a regression with higher-dimensional outputs. From empirical analysis, we find that polar prototype networks benefit from large margin separation and semantic class structure, while only requiring a minimal amount of output dimensions. While the structure is simple, the performance is on par with (classification) or better than (regression) standard network methods. Moreover, we show that we gain the ability to perform regression and classification jointly in the same space, which is disentangled and interpretable by design.

## 1 INTRODUCTION

This paper strives for classification *and* regression in neural networks. Current standards in classification (with softmax cross-entropy) and regression (with mean squared error) yield effective performance, but do so in a parametric manner. They do not use inductive biases such as maximum class separation, minimal description length, and domain knowledge (Mitchell, 1980). The awareness of such inductive biases then need to be learned while optimizing the task at hand. Instead, we propose to structure the network output space before learning, such that the class structure is both as simple as possible and maximally separable.

We are inspired by prototype-based networks, which divide an output metric space into Voronoi cells around a prototype per class (Guerriero et al., 2018; Hasnat et al., 2017; Jetley et al., 2015; Snell et al., 2017; Wen et al., 2016). The Voronoi division tackles the notion of simplicity and minimal description length, while enabling fast generalization and learning from new classes (Snell et al., 2017). However, by mapping examples to prototypes during training, and defining the prototypes as the mean of the examples, the output space is continually modified, altering the true prototype location. Obtaining prototypes is computationally expensive, as it requires a full pass over the training data. As a result, current prototype networks either focus exclusively on the few-shot classification setting (Snell et al., 2017) or resort to coarse approximations of the prototype locations (Hasnat et al., 2017; Guerriero et al., 2018). In this paper we question the need to learn prototypes.

By adopting a polar coordinate system, we show that classes can be positioned with (near) optimal separation *a priori*. This is achieved by finding an equal distribution of points on the hypersphere $\mathbb{S}^{D-1}$ for a $D$-dimensional output space. Fig. 1a shows an example for a 3D output space. By distributing classes based on maximal angular discrepancy, we arrive at an output space with large-margin class separation and an equal division of the output space over classes. Learning simplifies to minimizing the cosine distance between training examples and their class polar prototype. The structuring of the output space prior to learning alleviates the need to obtain and update the prototypes

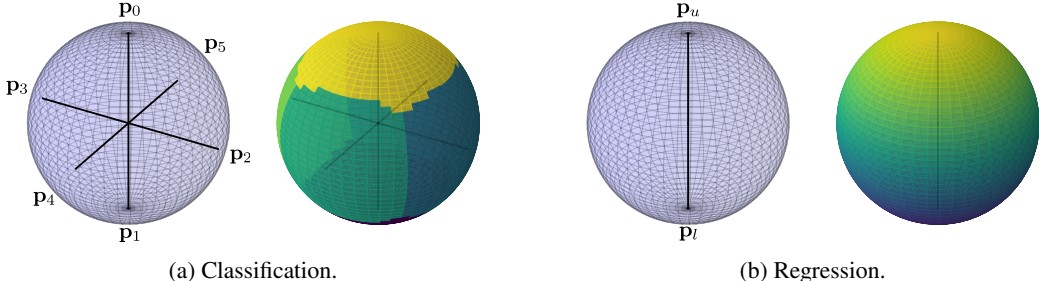

(a) Classification.               (b) Regression.

Figure 1: **Illustrative overview of polar prototype networks for classification and regression** with a 3-dimensional output space. For classification, we position class polar prototypes (six in Fig. 1a) in the output space with maximal separation prior to learning. The output space is determined by maximal polar similarity to the prototypes. For regression, we perform a polar interpolation of two opposing prototypes, denoting the lower and upper regression bounds ($\mathbf{p}_l$ and $\mathbf{p}_u$ in Fig. 1b). Note that the output space is visualized as a sphere for ease of visualization. We do not need to project output vectors explicitly onto the hypersphere during training or inference.

themselves. We consider two approaches to optimally distribute prototypes on the hypersphere. Furthermore, we investigate how to assign classes to prototypes by exploiting semantic priors.

Where the literature on prototype networks focuses exclusively on classification, we also propose a method for regression based on polar prototype networks (Fig. 1b). Two opposing polar prototypes are maintained for regression, denoting the lower and upper regression bounds. During training, we minimize for each example the difference between the expected and measured cosine similarities to the polar upper bound. Since the cosine similarities can be computed in higher-dimensional spaces, our approach is a direct generalization of standard regression, which learns on the one-dimensional line. Regression in polar prototype networks yields a more robust and better regression performance. What is more, as regression and classification now form a coherent unity, we show that in polar prototype networks we gain the ability to learn both tasks jointly within the *same* output, rather than separate outputs per task.

We make three contributions in this work: (*i*) we propose networks with simple, maximally separated, and semantic class structures in the output space defined prior to learning, (*ii*) we outline how to perform both classification and regression from the imposed output structures, and (*iii*) we show how to learn both tasks in the same output space in a disentangled and unified outputs. We will make the code and computed polar prototypes publicly available.

## 2 CLASSIFICATION AND REGRESSION WITH POLAR PROTOTYPES

### 2.1 CLASSIFICATION WITH POLAR PROTOTYPES

In a classification setup, we are given $N$ training examples $\{(\mathbf{x}_i, y_i)\}_{i=1}^N$, where $\mathbf{x}_i \in \mathbb{R}^I$ and $y_i \in C$ denote the inputs and class labels of the $i^{th}$ training example, $C = \{1, .., K\}$ the set of $K$ labels, and $I$ the input dimensionality. Furthermore, we have a set of $D$-dimensional polar prototypes $P = \{\mathbf{p}_1, ..., \mathbf{p}_K\}$, where each polar prototype $\mathbf{p}_k \in \mathbb{S}^{D-1}$ denotes a point on the $D$-dimensional hypersphere. The polar prototypes are maximally separated from each other *a priori*, *i.e.,* the prototypes provide an (approximately) equal separation of the output space. We first present the main loss function with the backpropagation and decision rule at inference. Then we outline how to obtain polar prototypes prior to learning and how to assign classes to prototypes.

#### 2.1.1 LOSS FUNCTION, BACKPROPAGATION, AND INFERENCE

For a training example $(\mathbf{x}_i, y_i)$, let $\mathbf{z}_i = f_\phi(\mathbf{x}_i)$ denote the $D$-dimensional output vector givenva network $f_\phi(\cdot)$. Since the output space is subdivided by the polar prototypes, we propose to train a classification network by minimizing the angle between the output vector and the polar prototype

$\mathbf{p}_{y_i}$ for ground truth label $y_i$, so that the classification loss $\mathcal{L}_{\text{polar-c}}$ to minimize is given as:

$$\mathcal{L}_{\text{polar-c}} = -\sum_{i=1}^{N} \cos\theta_{\mathbf{z}_i, \mathbf{p}_{y_i}} = -\sum_{i=1}^{N} \frac{|\mathbf{z}_i \cdot \mathbf{p}_{y_i}|}{||\mathbf{z}_i||\ ||\mathbf{p}_{y_i}||}. \tag{1}$$

The polar loss function yields a loss of minus one if the ouput vector and prototype point in the same direction. The loss increases as the angle becomes larger and equals one if the vectors point in opposite directions. We note that unlike common classification losses in deep networks, our loss function is only concerned with the mapping from training examples to a structured layout of the output space, the space itself does not need to be learned.

Since the polar prototypes do not require updating, we only have to backpropagate the error with respect to the training examples. The partial derivative of the loss function of Eq. 1 with respect to $\mathbf{z}_i$ is given as:

$$\begin{aligned}
\frac{d}{d\mathbf{z}_i} - \cos\theta_{\mathbf{z}_i, \mathbf{p}_{y_i}} &= -\frac{\mathbf{p}_{y_i} \cdot (||\mathbf{z}_i|| \cdot ||\mathbf{p}_{y_i}||) - \mathbf{z}_i \cdot |\mathbf{z}_i \cdot \mathbf{p}_{y_i}| \cdot ||\mathbf{p}_{y_i}|| \cdot ||\mathbf{z}_i||^{-1}}{||\mathbf{z}_i||^2 \cdot ||\mathbf{p}_{y_i}||^2} \\
&= \frac{\cos\theta_{\mathbf{z}_i, \mathbf{p}_{y_i}} \cdot \mathbf{z}_i}{||\mathbf{z}_i||^2} - \frac{\mathbf{p}_{y_i}}{||\mathbf{z}_i|| \cdot ||\mathbf{p}_{y_i}||}.
\end{aligned} \tag{2}$$

The remaining layers in the network are backpropagated in the conventional manner given the error backpropagation of the training examples of Eq. 2.

The network minimizes the angle between projected features and polar prototypes. For a new data point $\tilde{\mathbf{x}}$, the cosine similarity to all class prototypes is computed and the class with the highest similarity serves as the prediction:

$$c^* = \arg\max_{c \in C} \left( \cos\theta_{f_\phi(\tilde{\mathbf{x}}), \mathbf{p}_c} \right). \tag{3}$$

### 2.1.2 Obtaining and assigning polar prototypes

The loss function and corresponding optimization of polar prototype networks hinges on the presence of polar prototypes that separate the output space prior to learning. For $D$ output dimensions and $K$ classes, this amounts to a spherical code problem of optimally separating $K$ classes on the $D$-dimensional unit-hypersphere $\mathbb{S}^{D-1}$ (Saff & Kuijlaars, 1997). For $D = 2$, this can be easily solved by splitting the unit-circle $\mathbb{S}^1$ into equal slices, separated by an angle of $\frac{2\pi}{K}$. Then, for each angle $\psi$, the 2D coordinates are obtained as $(\cos\psi, \sin\psi)$.

For $D \geq 3$, no such optimal separation algorithm exists. This is known as the Tammes problem (Tammes, 1930), for which exact solutions only exist for optimally distributing a handful of points on $\mathbb{S}^2$ and none for $\mathbb{S}^3$ and up (Musin & Tarasov, 2015). To obtain polar prototypes for any output dimension and number of classes, we observe that the optimal set of prototypes, $P^*$, is the one where the largest cosine similarity between two class prototypes $\mathbf{p}_i$, $\mathbf{p}_j$ from the set is minimized:

$$P^* = \arg\min_{P' \in \mathbb{P}} \left( \max_{(k,l,k \neq l) \in C} \cos\theta_{(\mathbf{p}'_k, \mathbf{p}'_l)} \right), \tag{4}$$

where $\mathbb{P}$ contains all sets of $K$ vectors on $\mathbb{S}^{D-1}$.

We consider two approaches to approximating the objective of Eq. 4. The first employs Monte Carlo sampling, where a multifold ($1e7$) of polar prototype sets are sampled from a uniform distribution and the set which maximizes the objective is maintained. The second is an evolutionary algorithm (Eiben et al., 2003) that optimizes for polar prototype separation. For a set of class polar prototypes $P$, the evolutionary algorithm has a fitness function $g(P)$, which returns the minimum cosine distance between the pairs of prototypes in the individual. For every new generation (300 in total), we sample parents (30% from population of 3,000) using fitness proportionate selection and offsprings are produced using single-point row crossover (each parent pair produces 300 offsprings). Before insertion into the population, each new individual undergoes uniform mutation (each feature is replaced by uniform sample with $p = 0.01$). The final population size is decreased back to the original size by sampling individuals for survival, again proportional to $g(P)$.

Both Monte Carlo sampling and the evolutionary algorithm yield a set of polar prototypes with a large margin separation. However, the algorithms do not specify which classes should be represented by which prototypes. We consider two ways for class to prototype assignment. The first is to assign classes randomly to polar prototypes. The second is to structure the polar prototypes such that semantically similar classes point in similar directions. To achieve this we adapt our evolutionary algorithm to exploit word2vec (Mikolov et al., 2013) representations of the class names. To encourage finding polar prototypes that incorporate semantic information, we add a similarity score to the fitness function of the algorithm. This similarity score describes how similar the neighbourhoods of classes in a set of prototypes are to the neighbourhoods in the word2vec representations. For every class prototype $\mathbf{p}_i$, sorting the class prototypes based on the cosine similarity of both polar prototypes induces a ranking of 'closeness' to each other prototype $\mathbf{p}_j$. To compute the rank-based distance between the word2vec representations and the found polar prototypes, we can compute the distance between the word2vec order $w_{ij}$ and the ranks induced by the polar prototype order $p_{ij}$ as: $d_r(w, p) = \sum_{i,j} |\text{rank}(w_{ij}) - \text{rank}(p_{ij})|$. The similarity score is defined to be the inverse of $d_r(w, p)$ and is added to the fitness function with a weight parameter $\lambda = 10$ that allows us to balance the importance of the semantics and of the separation. Since the best ordering is given by the word2vec representations, we use Principal Component Analysis to reduce these to the desired number of dimensions and use the resulting representations to initialize our population. This evolutionary algorithm results in a set of polars with large margin separation and with a semantic structure from prior knowledge.

## 2.2 Regression with polar prototypes

While current prototype-based works focus exclusively on classification, we show here that regression can be naturally handled in polar prototype networks as well.

In a regression setup, we are given $N$ training examples $\{(\mathbf{x}_i, y_i)\}_{i=1}^{N}$, where $y_i \in \mathbb{R}$ now denotes a real-valued regression value. The upper and lower bounds on the regression task are denoted as $v_u$ and $v_l$ respectively and are typically the maximum and minimum regression values of the training examples. To perform regression with polar prototypes, we first observe that training examples should no longer point towards a specific prototype as done in classification. Rather, we posit that for regression we maintain two prototypes: $\mathbf{p}_u \in \mathbb{S}^{D-1}$ which denotes the regression upper bound and $\mathbf{p}_l \in \mathbb{S}^{D-1}$ which denotes the lower bound. Their specific direction is irrelevant, as long as the two prototypes are diametrically opposed, *i.e.,* $\cos \theta_{\mathbf{p}_l, \mathbf{p}_u} = -1$. The idea behind polar prototype regression is to perform a polar-based interpolation between the lower and upper prototypes. We propose the following loss function for regression with polar prototypes:

$$\mathcal{L}_{\text{polar-r}} = \sum_{i=1}^{N} (r_i - \cos \theta_{\mathbf{z}_i, \mathbf{p}_u})^2, \quad (5)$$

$$r_i = 2 \cdot \frac{y_i - v_l}{v_u - v_l} - 1. \quad (6)$$

The loss function of Eq. 5 computes a squared loss between two values. The first value denotes the ground truth

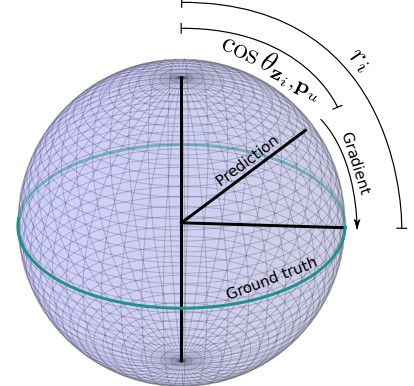

Figure 2: **Illustration of polar prototype networks for regression** with 3D outputs. For a training example, we compute the cosine similarity between the output prediction ($\mathbf{z}_i$) and polar upper bound ($\mathbf{p}_u$). This value is compared to the expected similarity ($r_i$) and difference between the values determines the gradient direction and magnitude.

regression value, normalized based on the upper and lower bounds. The second value denotes the cosine similarity between the output vector of the training example and the polar upper bound. The intuition behind the loss function is shown in Figure 2 for 3D output space. Shown is an artificial training example for which the ground truth regression value $r_i$ is zero. Due to the symmetric nature of the cosine similarity with respect to the polar upper bound, any output prediction of the training example on the turquoise circle is equally correct. As such, the loss function of Eq. 5 adjusts the angle of the output prediction either away or towards the polar upper bound, based on the difference between the expected and measured cosine similarity to the polar upper bound.

| | Semantic | CIFAR-10 | | | CIFAR-100 | | |
|---|---|---|---|---|---|---|---|
| | | Separation min | max | Accuracy | Separation min | max | Accuracy |
| **_Baseline prototypes_** | | | | | | | |
| One-hot vectors (Chintala et al., 2017) | | 1.00 | 1.00 | $91.1 \pm 0.2$ | 1.00 | 1.00 | $51.5 \pm 0.4$ |
| word2vec (Mikolov et al., 2013) | ✓ | 0.72 | 1.40 | $91.0 \pm 0.2$ | 0.26 | 1.32 | $61.2 \pm 0.7$ |
| **_Our prototypes_** | | | | | | | |
| Monte Carlo | | 0.77 | 1.76 | $90.9 \pm 0.1$ | 0.73 | 1.36 | $64.9 \pm 0.1$ |
| Evolutionary | | 0.89 | 1.81 | $90.7 \pm 0.2$ | 0.73 | 1.40 | $64.8 \pm 0.3$ |
| Evolutionary | ✓ | 0.91 | 1.40 | $91.2 \pm 0.1$ | 0.73 | 1.36 | $65.0 \pm 0.3$ |

Table 1: **The effect of maximum margin separation and semantic priors** for classification in polar prototype networks. Separation is quantified by the minimum and maximum cosine distance among the polar prototypes. We find that an approximate maximum margin separation using our Monte Carlo or evolutionary algorithm is sufficient for effective classification. Furthermore, we find that incorporating semantic priors in our evolutionary algorithms results in polar prototypes with large margin separation, semantic structure, and effective classification performance.

Standard regression computes and backpropagates a loss directly on one-dimensional outputs. In the context of this work, this corresponds to an optimization on the line from $\mathbf{p}_l$ to $\mathbf{p}_u$. Our approach generalizes regression to higher dimensional output spaces. While we still try to find an interpolation between two points on the line, the ability to project to higher dimensional outputs provides additional degrees of freedom to help the regression optimization. As shown in the experiments, this generalization results in a better and more robust performance than mean squared error.

### 2.3 LEARNING MULTIPLE TASKS IN THE SAME OUTPUT SPACE

To underline that classification and regression form a coherent unity in polar prototype networks, we show that both tasks can be optimized not only with the same base network, as is common in multi-task learning (Caruana, 1997), but can even be done in the same output space. To combine regression and classification in a shared $D$-dimensional output space, all that is required is to place the upper and lower polar bounds for regression in opposite direction along one axis. The other axes can then be used to maximally separate the class polar prototypes for classification. Optimization is as simple as summing the losses of Eq. 1 and 5. In this manner the regression and classification are unraveled and interpretable by design, while being jointly optimized in the same space. This allows us to obtain classification and regression results at the same time and to visualize and interpret results from multiple tasks in the same space.

## 3 EXPERIMENTAL EVALUATION

### 3.1 CLASSIFICATION

We first evaluate polar prototype networks for classification, where we investigate the importance of the inductive and semantic priors that are incorporated in our network outputs: maximum margin separation, semantic class similarity, and minimum description length. We evaluate on CIFAR-10 and CIFAR-100 using ResNet32 (He et al., 2016) as the base network, optimized using SGD (learning rate: 0.01, momentum: 0.9) for 250 epochs. The learning rate is decreased by a factor 10 after 150 and 200 epochs. For all experimental settings, we perform five runs and report the mean and standard deviation of the top-1 accuracy.

**Maximum margin separation.** First, we investigate the importance of separation with polar prototypes. We evaluate our Monte Carlo and evolutionary algorithms, with a comparison to baseline prototypes as proposed in (Chintala et al., 2017). In the baseline each prototype is a vertex of the standard simplex, *i.e.,* each prototype is a unique one-hot vector where all entries are zero except one. The baseline prototypes are optimally separated but on the positive quadrant only, while our prototypes are approximately optimally separated on the full hypersphere. For all approaches, we perform random assignments between classes and polar prototypes. Using these

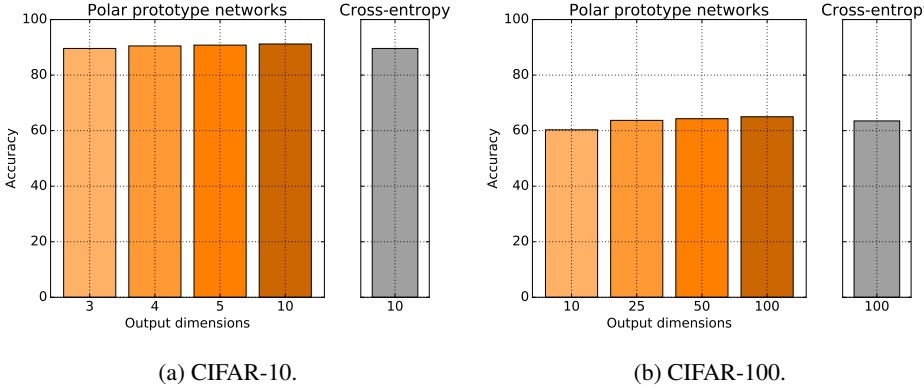

(a) CIFAR-10.                              (b) CIFAR-100.

Figure 3: **Minimal amount of output dimensions** in polar prototype networks. For CIFAR-10 and CIFAR-100, we can largely reduce the output space with minimal loss in performance. We also compare to softmax cross-entropy and find that the performance is similar. This experiment highlights the ability of our approach to provide effective results with the simplest of output structures.

three approaches, we relate the separation to the classification performance in polar prototype networks. For this experiment, we use 10 and 100 output dimensions for CIFAR-10 and CIFAR-100 respectively.

The results are shown in Table 1, where the approaches of interest are the non-semantic ones. We observe that the baseline has an equal separation among all classes, as the prototypes form an orthonormal basis. However, this also means only a quarter of the space of the hypersphere is used. Our approaches make full use of the space of the hypersphere. We find that the evolutionary algorithm yields a better separation than the Monte Carlo algorithm, especially in lower dimensions. The importance of exploiting the full hypersphere for polar prototypes becomes evident when evaluating the classification performance. While the CIFAR-10 performance is roughly equal for all three, the baseline only reaches an average accuracy of 51.5% on CIFAR-100, compared to 64.9% and 64.8% for our polar prototypes. Interestingly, the larger margins of our evolutionary algorithm do not result in better performance. These results indicate that for classification, a large margin separation on the full hypersphere is preferred, but a maximum separation is not a must.

**Learning with semantic priors.** Second, we investigate the effect of enhancing our evolutionary algorithm with semantic class knowledge. We use the word2vec embeddings of the class names as the baseline approach for semantic polar prototypes. The word embeddings are a natural baseline, as the cosine similarity is the common similarity measure for the embedding vectors. We perform PCA reduction on the word embeddings to make them of equal dimensionality as our polar prototypes. The results of this

|  | Samples per class | | | |
|---|---|---|---|---|
|  | 5 | 20 | 50 | all |
| Ours (w/o semantics) | 8.3 | 16.4 | 28.7 | 64.8 |
| Ours (w/ semantics) | 8.9 | 17.7 | 30.4 | 65.0 |

Table 2: **Classification for few training examples** on CIFAR-100. Adding semantic priors to our polar prototypes aids classification performance when examples are more limited.

experiment are in Table 1, with the semantic approaches check-marked. We observe that the word2vec baseline yields a desirable separation and accuracy on CIFAR-10. However, when increasing the output dimensionality and number of classes in CIFAR-100 the separation decreases, which has a negative effect on the performance (mean accuracy of 61.2%). For our evolutionary algorithm, we obtain polar prototypes with both a semantic structure and large class separation. The performance on CIFAR-10 and CIFAR-100 benefits from the semantics, leading us to conclude that semantic priors aid the classification abilities. In Table 2, we also investigate the use of semantic priors in polar prototype networks when training examples are scarce. When training examples are scarce our approach benefits from the semantic class correlations for classification. To show

| | | OmniArt | |
|---|---|---|---|
| | *Output dimensions* | *Optimizer* | |
| | | SGD | Adam |
| ***Baseline regression*** | | | |
| (Strezoski & Worring, 2017) | 1 | 88.3 | - |
| Square loss regression | 1 | $199.7 \pm 31.7$ | $81.0 \pm 1.4$ |
| Square loss regression | 2+1 | $236.2 \pm 74.3$ | $78.7 \pm 0.7$ |
| Square loss regression | 3+1 | $207.2 \pm 35.8$ | $80.0 \pm 0.6$ |
| ***Our regression*** | | | |
| Polar prototypes | 2 | $74.8 \pm 6.8$ | $71.9 \pm 3.6$ |
| Polar prototypes | 3 | $\mathbf{72.3 \pm 3.9}$ | $\mathbf{71.6 \pm 2.7}$ |

Table 3: **Regression performance of polar prototype networks** for predicting creation year in paintings. Shown are the mean absolute error rates. Polar prototype networks largely outperform standard network regression when using SGD. With Adam, we also report better performance. Polar prototype networks are an effective and robust solution for regression in neural networks.

that our approach also benefits from deeper architectures, we have also applied our approach with semantic priors on a DenseNet121 architecture (Huang et al., 2017). This improved the results to 93.2% (CIFAR-10) and 73.1% (CIFAR-100). We expect further improvements with strategies such as pre-training and extensive hyperparameter tuning.

**Towards minimal amount of dimensions in the network outputs.** Where the penultimate layer of networks with softmax cross-entropy is described by an unbounded $K$-dimensional output space for $K$ classes, our polar prototype networks reduces the description to a bounded $(K \times K)$-dimensional matrix when using the same number of output dimensions. Contrary to softmax cross-entropy however, we are not restricted to a fixed-size output space. In the third study, we evaluate the classification performance when using lower-dimensional output spaces, to arrive at a minimal amount of dimensions in the output space. We use our evolutionary algorithm with semantic priors. We also use this study to compare to softmax cross-entropy.

Figure 3 shows the classification performance as a function of the output description length on CIFAR-10 and CIFAR-100. For both datasets, we observe that we can reduce the output space for a large part with minimal effect on the performance. We also provide a comparison to softmax cross-entropy with identical settings, which yields a similar performance on both datasets. Next to being competitive to softmax cross-entropy, our approach is also competitive to the prototype-based approach of Guerriero et al. (2018), with the benefit that our approach not longer needs to continuously re-esitmate and update the class prototypes. A direct comparison to e.g. Schnell et al. (2017) is not possible, since it is optimized for few-shot learning only. We conclude that polar prototype networks deliver effective performance with the simplest of output structures, embodying Occam's Razor in enabling predictive interpretation (Grünwald et al., 2005).

## 3.2 REGRESSION

In the second experiment, we evaluate polar prototype networks for regression. We perform an evaluation on the challenging task of predicting the creation year of paintings. We focus on paintings from the $20^{th}$ century available as part of the large-scale OmniArt dataset (Strezoski & Worring, 2017). This results in a dataset with 15,000 training examples and 8,353 test examples[1]. We employ a ResNet16 network architecture (He et al., 2016) trained in similar fashion to the classification setup. The Mean Absolute Error is used as the evaluation metric. We compare our approach to a squared loss regression baseline, where we both normalize and clamp the outputs between 0 and 1 using the bounds to provide a fair comparison to our regression. For this baseline, we also include variants where the output layer has more dimensions, followed by an additional layer to a one-dimensional output.

---

[1]The dataset, regression labels, and train/test split will be made available online to enable reproduction.

The results are shown in Table 3. When we employ the same setup as in our classification experiments, *i.e.,* with SGD as the optimizer, the squared loss regression baselines all fail to converge, resulting in high error rates. Our approach using SGD does converge and yields far better results. Given the large difference in performance, we also ran the baselines and our approaches using Adam (Kingma & Ba, 2014). With this setting, the baselines do converge. However, our approach also works better. We find that, using Adam, our approach with a two-dimensional output significantly outperforms the baseline with a two-dimensional plus an additional one-dimensional layer ($p = 0.02$ for a two sample t-test, $H_0$ = both samples have same mean). This also holds for the comparison with three dimensions ($p = 0.01$). With our approach, we observe that using three dimensions over two results in slightly better performance, but not significantly ($p = 0.9$). Lastly, we find that our error rate of 71.6 outperforms the regression baseline of (Strezoski & Worring, 2017) (MAE of 88.3, other info on paintings is excluded for a more fair comparison to our work). We conclude that polar prototype networks provide an effective and robust solution for regression.

### 3.3 JOINT REGRESSION AND CLASSIFICATION IN THE SAME OUTPUT SPACE

To underline the unification of regression and classification in polar prototype networks, we have performed an experiment where both tasks are optimized within the same output space. We use the MNIST dataset, where we aim to both classify the digits and regress the rotation of the examples. We use the digits 2, 3, 4, 5, and 7, where we apply a random rotation between 0 and 180 degrees to each example. The other digits were not of interest given the rotational range. We employ a 3-dimensional output space, where the classes are optimally separated along the $(x, y)$ plane and the regression bounds are projected along the $z$-axis. We use a simple network with two convolutional and two fully connected layers.

In Figure 4, we visualize projections of data points in the output space of our network after 20 epochs. We provide separate visualizations for regression and classification for clarity and emphasize that they are the same output space. For regression, we observe a smooth interpolation between the minimum and maximum rotation, highlighting our regression ability. For classification, we observe that the digits from each class form slices with large separation to other classes. The joint optimization results in a space that unravels the inter-class separation from the intra-class rotations by design.

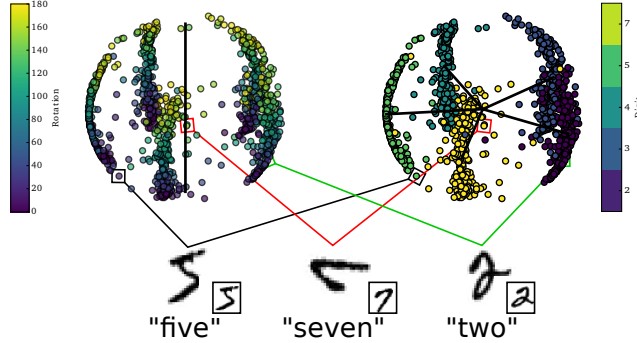

Figure 4: **Joint regression and classification** in the same output space on a rotated MNIST subset. The regression and classification structure is shown separately for ease of visualization, we explicitly note that the space is identical. Rotation is along the $z$-axis (color denotes regression value), while classification is on the $(x, y)$-plane (color denotes class). With polar prototype networks, the output space can disentangle different tasks in a structured manner.

## 4 RELATED WORK

Our approach builds upon prototype-based networks, which have recently gained traction under various names, including centers (Wen et al., 2016), proxies (Movshovitz-Attias et al., 2017), means (Guerriero et al., 2018), prototypical concepts (Jetley et al., 2015), and prototypes (Snell et al., 2017). In general, these works adhere to the Nearest Mean Classifier paradigm (Mensink et al., 2013) by assigning training examples to a vector in the output space of the network, which is defined to be the mean vector of the training examples. A few works have also investigated multiple prototypes per class (Movshovitz-Attias et al., 2017; Yang et al., 2018). Prototype-based networks have shown to enforce a more coherent output structure (Wen et al., 2016) and to enable a generalization to new classes (Guerriero et al., 2018; Snell et al., 2017; Yang et al., 2018).

While promising, the training of prototype networks is currently faced with a chicken-and-egg problem. Training examples are mapped to class prototypes, while class prototypes are defined as the mean of the training examples. Because the projection from input to output changes continuously during network training, the true location of the prototypes changes with each mini-batch update. This holds for methods with both one, or multiple prototypes per class. Obtaining the true location of the prototypes is expensive, as it requires a pass over the complete dataset. As such, prototype networks currently either focus on the few-shot setting (Boney & Ilin, 2017; Snell et al., 2017), or on approximating the prototypes. Approximations include alternating the example mapping and prototype learning (Hasnat et al., 2017) or updating the prototypes online as a function of the mini-batches (Guerriero et al., 2018). We propose to bypass the prototype learning altogether by structuring the output space prior to training. By defining polar prototypes as points on the hypersphere, they are maximally separated *a priori*. Optimization simplifies to minimizing a polar distance between training examples and their corresponding prototype, alleviating the need to continuously obtain and learn prototypes. Moreover, we provide a unification between classification and regression.

Bojanowski and Joulin (Bojanowski & Joulin, 2017) recently showed that unsupervised learning is possible by projecting examples to random prototypes on the unit hypersphere. Here, we similarly employ prototypes on the hypersphere, but do so in a supervised setting, without the need to explicitly project network outputs to the hypersphere. Perrot and Habrard (Perrot & Habrard, 2015) have previously explored the notion of pre-defined prototypes in the context of metric learning. We also employ pre-defined prototypes, but on the hypersphere in a deep learning setting using polar distances, for both classification and regression.

Several works have shown the benefit of optimizing networks using angles for classification. Liu *et al.* (2016) aim to improve the separation in softmax cross-entropy by increasing the angular margin between classes. In similar fashion, several works project network outputs to the hypersphere for classification through $\ell_2$ normalization, which forces softmax cross-entropy to optimize for angular separation (Hasnat et al., 2017; Liu et al., 2017a; Wang et al., 2018; Zheng et al., 2018). The work of (Gidaris & Komodakis, 2018) shows that using cosine similarity in the output helps the generalization to new categories. The potential of angular similarities has also been investigated in other layers of deep networks (Liu et al., 2017b; Luo et al., 2017). In this work, we combine large margin separation from an angle-based perspective with the minimal description length from a prototype-based perspective and arrive at polar prototype networks, a simple and effective unified solution for classification and regression.

## 5 CONCLUSIONS

We propose polar prototype networks for unified classification and regression by imposing output structures that are both simple and with maximum margin separation. For classification, prototypes are distributed as uniformly as possible on the hypersphere prior to learning. We also show how to incorporate a semantic structure in the prototype distribution from prior knowledge. The network in turn only requires a polar minimization between training examples and their class prototypes. The simplicity of the output structure furthermore enables us to generalize to regression. Rather than directly minimizing an angle to fixed prototype locations, as done in classification, we maintain two diametrically opposed prototypes; the lower and upper regression bound. We outline how to perform regression in this setting as an interpolation between the polar lower and upper bounds. Experimentally, we show that our classification approach benefits from large-margin class separation and a semantic structure, resulting in effective classifiers with a minimal description length. Our regression outperforms standard squared error optimization, highlighting its expressive abilities. Since regression and classification are unified in the same network, we enable the capability to optimize both tasks in the same output, resulting in a common space with disentangled factors. We conclude that polar prototype networks provide a unified solution at least as effective as softmax cross-entropy and mean squared error, while having simpler and more interpretable output structures.

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
