# OpenReview forum: "Polar Prototype Networks"
_ICLR.cc/2019/Conference_

### Official Review · AnonReviewer2 · 2018-10-16
**An interesting paper, but still need to improve**

**Rating:** 4
**Confidence:** 5

**Review:**

This paper unifies both classification and regression task based on the polar prototype network. For classification, the prototypes for all classes are chosen in advance based on a max-margin principle, while the embedding of all instances is then optimized to have small cosine distance to assigned prototypes. For the regression, the output value is interpolated between the two prototypes. Experiments on classification, regression, and combined tasks show the method can achieve good results.

The idea of using the prototype and the polar system is interesting, and the whole paper is well-written. However, there are still some problems and questions about this paper.
1. There are two problems with using the max-margin prototypes. First, to maximize the smallest distance between two prototypes, the authors use MC or evolutionary algorithms to do the optimization, which may be time-consuming, and it may be extremely difficult when the prototype space is high dimension. Second, the previous approach indeed obtains discriminative prototypes, but we lose the **class correlation**. In the extreme case, it is equal distance between all prototypes, but some similar classes will have a smaller prototype distance than others. For example, the prototype distance between "cat" and "dog" should not be the same as that between "car1" and "car2". The semantic consideration in the paper can solve this problem to some extent, but there needs more evidence.

Using the pre-defined prototype is also considered in the paper "M. Perrot et al. Regressive Virtual Metric Learning. NIPS15".

2. For the unified output space
One main contribution is that based on the polar system, the method unifies both classification and regression tasks in the same space. We can also do this in basic embedding algorithms. In the embedding space, a method can do both classification and regression with the nearest neighbor rule (based on majority voting and average respectively). The authors should compare with such kinds of methods in the experiments.

3. Experiments
From the experiments, using semantic cannot improve a lot for the classification task. The authors can try more datasets to validate is this the common scenario. The reviewer strongly suggests the authors should compare with more methods. For example, in some papers the prototypes are learned simultaneously (Snell et al. Prototypical networks for few-shot learning. NIPS17; Wen et al. A discriminative feature learning approach
for deep face recognition. ECCV16); while in other cases, there are no prototypes as we optimize the triplet/contrastive loss directly. First, the authors can compare classification performance with these approaches; besides, some visualization results can also show the used prototypes or embeddings.
The main advantage of the method is not stressed clearly in the experiments part. The authors can clarify it in later versions.

The final rating depends on the authors' response.

---

> ### Author Response · Authors · 2018-11-25
> **Rebuttal to reviewer**
>
> We thank the reviewer for the comments and suggestions to improve the paper and for deeming the paper interesting and well written.
>
> Regarding the max-margin prototypes, we agree that computing maximally separating polar prototypes is difficult and time-consuming. We note that we only have to compute the polar prototype once off-line for each combination of class size and number of output dimensions. We have readily computed polar prototypes for multiple output dimensions and number of classes (corresponding to many popular datasets) and we will make these prototypes publicly available to avoid recomputing them. For a desired new combination of classes and number of output dimensions, it takes in the order of a few minutes to compute polar prototypes once.
>
> Regarding the class correlation in max-margin polar prototypes, we agree that intuitively, not all classes should be equally well separated. Incorporating semantic when computing polar prototypes can solve this problem to some extent, as noted by the reviewer. To further convince the reviewer that our polar prototypes with semantic consideration indeed incorporate class correlation and that this information helps the classification performance, we have performed an experiment where we investigate the classification performance of polar prototypes as a function of the number of training examples. We have investigated the performance of polar prototypes as a function of the number of training examples. Below, we show the results on CIFAR-100 for multiple numbers of training examples per class for our approach both with and without class correlations. The results show that incorporating semantics can help the classification performance when training examples are scarce, especially between 20 and 50 examples per class. We will add the table and discussion to Section 3.1 of the paper.
>
> -------------------------------------------------
> NR. EXAMPLES PER CLASS | 5   | 20   | 50   | all
> -------------------------------------------------
> OURS (NON-SEMANTIC)    | 8.3 | 16.4 | 28.7 | 64.8
> OURS (SEMANTIC)        | 8.9 | 17.7 | 30.4 | 65.0
> -------------------------------------------------
>
> We thank the reviewer for pointing out the reference to the NIPS 2015 paper using pre-defined prototypes. We have added the following discussion to the related work:
> "Perrot and Habrard [1] have previously explored the notion of pre-defined prototypes in the context of metric learning. We also employ pre-defined prototypes, but on the hypersphere in a deep learning setting using polar distances, for both classification and regression."
>
> We agree with the reviewer that it is also possible to have a joint embedding space for classification and regression. The tasks are however not unified in standard embedding spaces, since they are still optimized differently. We show in Section 3.3 and Figure 4 that our approach enables a joint space that can be directly visualized and interpreted, due to the explicit disentanglement of the dimensions.
>
> We have added a comparison and discussion to the prototype-based approach of Guerriero et al. [2] to Section 3.1 of the paper. This approach yields results roughly similar to softmax cross-entropy, as is also the case with our approach. The work of Guerriero et al. is however limited in two aspects compared to our approach. First, a continuous update of the class prototypes is required, as the exact location changes with every network weight update. This makes the network less efficient to optimize. Second, the approach performs classification only, while our approach unifies classification and regression under the same umbrella. We note that a comparison to prototype-based approaches such as Schnell et al. [3] are not directly possible, as they are limited to the few-shot scenario only (given their requirement for all examples of a class in each mini-batch). This point will be clarified in Section 3.1 of the paper.
>
> [1] Perrot, Michaël, and Amaury Habrard. "Regressive virtual metric learning." NIPS. 2015.
> [2] Samantha Guerriero, et al. "Deep nearest class mean classifiers." ICLR, Worskhop Track. 2018.
> [3] Jake Snell, et al. "Prototypical networks for few-shot learning." NIPS. 2017.

---

### Official Review · AnonReviewer1 · 2018-10-29

**Rating:** 3
**Confidence:** 4

**Review:**

Proposal to use polar regression for prediction problems. To do so, one maps the target variable into "maximally separating prototypes" laid in the D-hypersphere. For classification, the learning problem reduces to minimizing the angle between D-dimensional feature vectors and the associated D-dimensional polar prototype. A similar strategy applies to regression, where the continuous target variable is squeezed to the range of the hypersphere.

The authors claim that their method unifies, as opposed to much prior art, classification and regression approaches. I disagree with this claim, since we usually approach classification as a (normalized!) regression problem. In some cases the normalization is on the entire output space (single-label classification as in ImageNET), and in some other cases this normalization happens separately in each component of the output space (multi-label classification as in COCO). It is even possible to train an ImageNET classifier using mean squared error given unit-norm feature vectors (Tygert et al, 2017). As such, the "unification" proposed by the paper seems a bit blurry to me.

I am unconvinced about the impact shown by the experiments. Table 1 shows accuracies far from the state-of-the-art (91% for all methods in CIFAR-10 versus 97% SOTA, 65% for the proposed method versus 75% SOTA) and throw some separation statistics without a clear correlation to accuracy. The experiment on semantic priors is inconclusive, as all non-baseline results are within error bars. The impact of Section 3.3. is also unclear, since obtaining semantic (digit rotation) interpolations in MNIST is a common feat achieved by unsupervised learning algorithms with decent feature learning.

The results from section 3.2 are interesting, although I would be interested in seeing a reduction-to-classification baseline, where the years are clustered to set up a classification problem, and the prediction is fine-tuned by a local regression.

Note: Regressing to (random) polar prototypes was proposed in https://arxiv.org/abs/1704.05310

---

> ### Author Response · Authors · 2018-11-25
> **Rebuttal to reviewer**
>
> We thank the reviewer for the comments to improve the paper and we would like take this opportunity to discuss the raised concerns.
>
> Regarding the unification between regression and classification, we agree that in standard network settings, classification can be seen as a normalized regression variant. This holds however only for the forward pass of the network. The loss functions and error backpropagation of classification (e.g. softmax/sigmoid cross-entropy) and regression (e.g. squared loss) are completely disjoint. In this work, we propose a network that unifies both approaches both in the forward and backward pass of the network. As pointed out by the reviewer, Chintala et al. [1] already apply squared loss on normalized output vectors for classification. We note two things however: (i) our polar prototypes outperform the prototypes of Chintala et al. when using our polar prototype networks and (ii) Chintala et al. do not show whether and how their approach generalizes to standard regression.
>
> Regarding the classification performance, the main goal of our experiments was to show that our polar prototypes outperform baseline polar prototypes, we are competitive with softmax cross-entropy, and we only need a minimal amount of dimensions in the output space. For the initial submission, our goal was not a state-of-the-art comparison and we have therefore employed standard hyperparameter settings and a standard network architecture (ResNet32). To convince the reviewer that the performance reported in the paper is not a result of the method, we have also performed an experiment using a DenseNet architecture [2]. Using this architecture, our approach obtains a classification performance of 93.2% (CIFAR-10) and 73.1% (CIFAR-100). We expect further improvements when employing more elaborate tuning tactics such as pre-training and extensive hyperparameter tuning. We will add this experiment to Section 3.1.
>
> To further investigate the impact of semantic priors in classification, we have performed an experiment on CIFAR-100 that shows the classification performance as a function of the number of training examples per class. The results are shown in the Table below. We find that when training examples are scarce, semantic priors help to increase the classification performance, especially between 20 and 50 examples per class. We will add the Table to Section 3.1 of the paper.
>
> -------------------------------------------------
> NR. EXAMPLES PER CLASS | 5   | 20   | 50   | all
> -------------------------------------------------
> OURS (NON-SEMANTIC)    | 8.3 | 16.4 | 28.7 | 64.8
> OURS (SEMANTIC)        | 8.9 | 17.7 | 30.4 | 65.0
> -------------------------------------------------
>
> We thank the reviewer for the positive feedback regarding the evaluation in Section 3.2. The main goal of the experiment was to show that our approach provides a statistically significantly better performance than the de facto standard in regression, namely squared loss. We are currently looking into the additional regression baseline suggested by the reviewer and we hope to provide results soon that can be added to Table 2 of the paper.
>
> We agree with the reviewer that the results of Section 3.3 that the MNIST setting is not highly challenging, nor was this our goal. The goal of the final experiment was to highlight our ability to explicitly disentangle different tasks within the same output space. In this setting, the rotation regression was along the z-axis and the classification along the xy-plane. Figure 4 shows that in this setup, such a joint space is feasible. Such a joint space allows us to yield classification and regression result at the same time and allows us to quickly visualize the prediction in multiple tasks and more importantly, their interaction. For example, a mistake in one task can be explained by a mistake in the other task. We will make the point of unified output spaces for classification and regression more clear in Section 2.3.
>
> We thank the reviewer for pointing out the reference. We have added the following discussion to the related work:
> "Bojanowski and Joulin [3] recently showed that unsupervised learning is possible by projecting examples to random prototypes on the unit hypersphere. Here, we similarly employ prototypes on the hypersphere, but do so in a supervised setting, without the need to explicitly project network outputs to the hypersphere."
>
> [1] Soumith Chintala, et al. "Scale-invariant learning and convolutional networks." Applied and Computational Harmonic Analysis. 2017.
> [2] Huang, Gao, et al. "Densely connected convolutional networks." CVPR. 2017.
> [3] Bojanowski and Joulin "Unsupervised learning by predicting noise." ICML. 2017.

---

### Official Review · AnonReviewer3 · 2018-10-30
**Interesting idea but need more convincing experiments**

**Rating:** 5
**Confidence:** 3

**Review:**

This paper proposes a unified framework for both classification and regression and a combination of both using pre-designed prototypes distributed on a hypersphere with max separation. It is nice to see an alternative to the dominant cross-entropy loss and l2 loss for deep classification and regression respectively, also the ability to tackle both in a shared output space is a plus. However, the experiments are limited and not convincing that the proposed framework offers a genuine alternatives to existing formulations.

Pros:

•	The over idea appears to novel, despite its connections to various previous attempts to angular separation (Hasnat et al., 2017; Liu et al., 2017a; Wang et al., 2018; Zheng et al., 2018).
•	It is nice to see a framework that can perform classification/regression multi-task learning. Many computer vision problems have this nature, e.g. object recognition and pose estimation; face recognition and age estimation. So addressing both problems jointly can potentially bring in mutual benefits.

Cons:

•	The experiments on CIFAR 10/100 seems to be at par with a conventional cross-entropy loss. It would be more convincing if more experiments on other more challenging datasets (e.g. ImageNet) using more powerfully backbone networks (e.g. densenet) can be provided.
•	In the CIFAR experiments the hypersphere space dimension is set to the same as the number of classes. In this case, why not just use a one-hot vector to represent each class, and do L2 normalization to the output of the feature extraction network and then do softmax cross-entropy? As pointed out in Section 4, several works project network outputs to the hypersphere for classification through L2 normalization, which forces softmax cross-entropy to optimize for angular separation (Hasnat et al., 2017; Liu et al., 2017a; Wang et al., 2018; Zheng et al., 2018). These works should be compared here to convince the readers why it is necessary to use evolutionary algorithm or Monto-Carlo sampling to set the prototypes rather than just using one-hot vectors.

---

> ### Author Response · Authors · 2018-11-25
> **Rebuttal to reviewer**
>
> We thank the reviewer for the helpful comments and encouraging words regarding the novelty and potential benefit of the proposed approach.
>
> Regarding the classification experiments, we agree that more experiments on bigger datasets and/or bigger networks further highlight the potential of our approach. The main goal of our classification experiments was to show that (i) our polar prototypes outperform baseline polar prototypes, (ii) we are competitive with softmax cross-entropy, and (iii) our approach only requires a minimal amount of dimensions in the output space. As such, we have opted to employ conventional datasets (CIFAR-10 and CIFAR-100) with a conventional deep network (ResNet32) for our experiments.
>
> Based on the suggestion from the reviewer, we have investigated bigger datasets and networks. We have performed experiments on CIFAR-10 and CIFAR-100 using DenseNet [1]. Below, we show an overview of the results.
>
> ----------------------------------------------
> POLAR PROTOTYPES |  CIFAR-10  |  CIFAR-100
> ----------------------------------------------
> ResNet32         |    91.2    |     65.0
> DenseNet121      |    93.2    |     73.1
> ----------------------------------------------
>
> The results show that our approach using evolutionary polar prototypes benefits from bigger and deeper networks, by comparing to the earlier reported numbers of 91.2% (CIFAR-10) and 65.0% (CIFAR-100) in the paper. Using a DenseNet with 121 layers, we can further improve the classification results (93.2% and 73.1% on resp. CIFAR-10 and CIFAR-100). We expect further improvements with more tuning strategies such as pre-training and extensive hyperparameter evaluation. This result underlines that our approach obtains competitive results regardless of the specific network architecture choices. We have furthermore started experiments on ImageNet with evolutionary polar prototypes, where we expect similar results as the CIFAR-10 and CIFAR-100 experiments using ResNet and DenseNet. We will incorporate the DenseNet experiments on CIFAR-10 and CIFAR-100 to the first ablation study of Section 3.1. The ImageNet experiments will be added to Table 1 with a discussion in Section 3.1.
>
> We agree with the reviewer that a comparison to one-hot vectors for class polar prototypes is required to be more convincing that the Monte Carlo / Evolutionary algorithms are required in polar prototype networks. In the paper have opted to provide a direct as possible comparison to this baseline, by placing polar prototypes on the simplex, which amounts to each class being a one-hot vector in a unique dimension. For this baseline, we employ the same network and settings as with the Monte Carlo and Evolutionary polar prototypes. We found that on CIFAR-10, this one-hot vector baseline works roughly as well as our approach. On CIFAR-100 however, we found a significant drop in performance (51.5%+/-0.4% for the baseline vs. 64.9%+/-0.1% for the Monte Carlo prototypes, both using a standard ResNet architecture). This result shows that our polar prototypes are required for competitive performance, especially when the number of classes increases. We will rename the "Simplex" baseline prototype "One-hot vector" in Table 1 and Section 3.1 to make this baseline more clear.
>
> [1] Huang, Gao, et al. "Densely connected convolutional networks." CVPR. 2017.

---

### Author Response · Authors · 2018-11-25
**Overview of changes to the paper**

We thank the reviewers for their feedback and suggestions to improve the paper. We have provided a rebuttal for each reviewer separately below. Based on the reviews, we have made the following updates to the paper:

- We have added a classification experiment with improved performance using a deeper architecture, namely a DenseNet with 121 layers.
- We have added an additional experiment showing the performance of our approach with and without semantic priors using few samples per class on CIFAR-100, to make the inclusion of semantic class information more convincing.
- We have included a discussion to additional references in the related work.
- We have included a discussion and comparison to other prototype-based approaches to Section 3.1.
- We have provided clarifications throughout the paper based on the comments of the reviewers.

---

### Meta-Review · Area_Chair1 · 2018-12-13

**Confidence:** 5
**Recommendation:** Reject

**Metareview:**

This work proposes a class of neural networks that can jointly perform classification and regression in the output space.
The authors explore the concept of polar prototypes which are points on the hypersphere in the output space. For classification, each class is described by a single polar prototype and training is equivalent to minimizing angular distances between examples and their class prototypes. For regression, training can be performed as a polar interpolation between two prototypes.
As rightly acknowledged by R3, “it is nice to see an alternative to the dominant cross-entropy loss and l2 loss for deep classification and regression respectively, also the ability to tackle both” at the same time.

However, all reviewers and AC agreed that the current manuscript lacks convincing empirical evaluations that clearly show the benefits of the proposed approach. To strengthen the evaluation, (1) see R1’s concern regarding the state-of-the-art performance on CIFAR-10;  (2) see R3’s suggestion to use more challenging datasets (e.g. ImageNet), stronger backbone networks (e.g. densenet), and also other applications (e.g. object recognition and pose estimation; face recognition and age estimation as classification and regression problems); (3) see R2’s suggestions for more baselines to be compared to.
Two other requests to further strengthen the manuscript are: (1) finding alternative ways to MC or evolutionary algorithms (R2); (2) exploring class correlation in the prototype space (R2).

In the response, the authors acknowledged that their initial results were not aimed for state-of-the-art comparison, but to show that the proposed objective is comparable to minimizing softmax cross-entropy loss. The authors provide additional experiments using DenseNet as the base network and the results are still slightly inferior to state-of-the-art performance.
The experiments using ImageNet dataset have been promised by the authors (in response to R3), but are not included in the current revision.

AC suggests in its current state the manuscript is not ready for a publication. We hope the reviews are useful for improving and revising the paper.